# Tissue-Specific Transcriptomic Responses to Avian Reovirus Inoculation in Ovo

**DOI:** 10.3390/v17050646

**Published:** 2025-04-29

**Authors:** Zubair Khalid, Shahna Fathima, Ruediger Hauck

**Affiliations:** 1Department of Pathobiology, Auburn University, Auburn, AL 36849, USA; zzk0012@auburn.edu; 2Department of Poultry Science, Auburn University, Auburn, AL 36849, USA; szf0075@auburn.edu

**Keywords:** avian reovirus, transcriptome, embryo, viral replication, RNA-seq

## Abstract

Avian reovirus (ARV) infections significantly impact the global poultry industry, but host responses across infection models remain poorly characterized. Using specific-pathogen-free chicken embryos, this study examined tissue-specific transcriptomic changes following in ovo inoculation with two doses of ARV S1133 at embryonic day 18. Quantitative PCR confirmed dose- and time-dependent viral replication, with the liver exhibiting the highest viral load at 24 h post-inoculation (hpi), whereas the kidneys, intestines, and bursa were only positive at 48 hpi with the higher viral dose. Transcriptomic profiling revealed the intestines mounted an extensive gene expression response, implicating early immune activation. Liver samples demonstrated strong upregulation of antiviral pathways, including interferon signaling and viral replication inhibition, while kidneys and intestines were enriched for coagulation and wound healing pathways. The bursae exhibited minimal immunity-related responses, suggesting insufficient maturation. Functional analyses confirmed tissue-specific immune and metabolic adaptations to infection. These findings indicate that ARV replication efficiency and host molecular responses are dose-, tissue-, and time-dependent. Notably, intestinal responses suggest preemptive immune engagement, while hepatic antiviral mechanisms may play a critical role in restricting viral spread. This study establishes foundational knowledge of host molecular responses to ARV in late-stage embryos, with implications for in ovo vaccination and early immunity.

## 1. Introduction

Specific-pathogen-free (SPF) embryonated eggs serve as a useful model to study the pathogenesis of various avian pathogens, including avian reovirus (ARV). As a globally prevalent pathogen [1] with a significant economic impact [2], ARV has been demonstrated to induce strain- and inoculation-route-dependent pathology in embryos [3,4,5,6,7]. In embryos ranging from 6 to 11 days of embryonation (DOE), ARV inoculation in the yolk sac [5] and chorioallantoic membrane (CAM) [3,4,7] resulted in varying lesions, including stunted embryos with subcutaneous hemorrhages, greenish livers with necrotic foci, edematous CAMs presenting white pock-like lesions, and enlarged spleens. Although these lesions were observed at an early DOE, which is advantageous for virus isolation, later stages of embryonation are better suited to investigate the molecular responses to viral infections, as cellular differentiation and immune system maturation continue until and even after hatching [8,9]. This has been demonstrated by a more persistent and robust immune response to the Newcastle disease virus at 18 DOE compared to 10 and 14 DOE [10]. Furthermore, previous studies have confirmed specific antiviral immune responses in the spleens and thymi of embryos infected with either the classical or attenuated infectious bursal disease virus at 18 days of embryonic development (DOE) [11].

From an applied perspective, examining antiviral responses at 18 DOE is of significant relevance because of in ovo immunization attempts against ARV [12,13,14,15]. For instance, the administration of a commercial ARV vaccine in ovo has been shown to suppress post-hatch T cell-mediated immunity [14]. However, the molecular responses and mechanisms underlying immunosuppression following ARV infection or immunization in the embryo remain unexplored. Insights into these pathways could enhance our understanding of the embryo’s response to the vertical transmission of ARV, a phenomenon that has been experimentally validated [16,17], with evidence indicating ARV-positive embryo mortality occurring as late as 21 DOE [18]. While the transcriptome-wide gene expression patterns of fibroblast cells [19] and spleens from SPF chickens [20] infected with ARV have been studied, the organ-specific molecular response to ARV inoculation in embryos remains unexplored. Characterization of these responses in late-stage embryos could provide insights into the mechanisms by which a near-hatch chick responds to ARV infection.

This study aimed to assess viral replication and transcriptional dynamics in various embryonic organs at 24 and 48 h post-infection (hpi) following inoculation with ARV S1133 at 18 DOE. The findings suggest dose, time, and tissue-specific molecular responses to avian reovirus.

## 2. Materials and Methods

### 2.1. S1133 Inoculum Preparation and Dose Determination

The ARV S1133 strain from a previously prepared and sequenced stock [21] was passaged once in chicken embryo liver (CELi) cells. The harvested cell lysate was frozen–thawed three times to release the virus. The virus was titrated on CELi cells with 8 replicates per dilution, and the titer was calculated to be 10^9.33^ TCID50/mL, using the Reed and Muench method [22]. The stock was diluted in DMEM to prepare inoculum at a final concentration of 10^6^ TCID50/mL or 10^4^ TCID50/mL.

### 2.2. Embryo Inoculation and Sampling

Specific-pathogen-free (SPF) eggs were purchased from AVS Bio (Roanoke, IL, USA) and incubated under recommended environmentally controlled conditions until 18 days of embryonation (DOE). Live embryos were divided into three groups: control (NC; DMEM-inoculated, *n* = 14), and Low (*n* = 14), and High (*n* = 14) S1133 groups inoculated with 100 µL containing 10^4^ and 10^6^ TCID_50_/mL of the pathogenic strain ARV S1133, respectively.

After puncturing the eggshell, the inoculum was deposited into the allantoic cavity of the embryo using a one-inch 21-gauge needle fully inserted at a 45-degree angle. The control group was injected with 100 µL of DMEM. Following inoculation, eggs were sealed with Clear Washable School Glue (Elmer’s Products Inc., Westerville, OH, USA), returned to the incubator, candled, and monitored twice a day until sampling.

### 2.3. Sampling

No mortality was observed in any group after the injection of the respective inoculum. At 24 and 48 hpi, 7 embryos from each group were decapitated, and intestine, liver, kidney, and bursa of Fabricius samples were collected in DNA/RNA Shield (Zymo Research, Irvine, CA, USA). Following collection, the tissues in the reagent were allowed to absorb the preservation reagent at 4 °C for 48 h and then stored at −80 °C.

### 2.4. RNA Extraction and Viral Load Determination

RNA was extracted using the RNeasy Mini Kit (Qiagen, Hilden, Germany) as per the manufacturer’s instructions. Viral RNA load was quantified by targeting the M1 gene of ARV. Total RNA was denatured for 5 min at 90 °C, and RNA was reverse transcribed using the LunaScript RT SuperMix Kit (New England Biolabs, Ipswich, MA, USA). The qPCR was performed using Forget-Me-Not™ Universal Probe qPCR Master Mix (Biotium, Fremont, CA, USA) using the following primers: forward (5′-ATG GCC TMT CTA GCC ACA CCT G-3′), reverse (5′-CAA CGA RAT RGC ATC AAT AGT AC-3′), and probe (5′-FAM-TGC TAG GAG TCG GTT CTC GTA-BHQ1-3′) [23]. The GAPDH gene was used to normalize the viral loads, using the de novo designed primers as follows: forward (5′-TGGTGGCCATCAATGATCCC-3′), reverse (5′-ACCTGCATCTGCCCATTTGA-3′), and probe (5′-FAM-ACTGTCAAGGCTGAGAACGG-BHQ1-3′). The PCR reaction was performed under the following conditions: initial enzyme activation at 95 °C for 3 min, followed by 40 cycles of denaturation at 95 °C for 5 s, annealing at 60 °C for 30 s, and extension at 72 °C for 30 s. Fluorescence was measured at the end of each extension step. Each run included positive, negative, and no-template controls to ensure assay reliability. All thermal cycling was carried out using the qTOWER^3^ PCR Thermal Cycler (Analytik Jena, Jena, Germany). Amplification peaks were analyzed by the qPCRSoft program version 4.1. Relative viral RNA loads were calculated using the formula log (2^−(GAPDH Ct−ARV Ct)^). Statistical comparisons for tissue, time, and dose were performed using three-way ANOVA.

### 2.5. RNA Sequencing

RNA aliquots were submitted to Azenta Life Sciences (South Plainfield, NJ, USA) for library preparation and sequencing. Samples were quantified using a Qubit 2.0 Fluorometer (Life Technologies, Carlsbad, CA, USA), and RNA integrity was checked using Agilent TapeStation 4200 (Agilent Technologies, Palo Alto, CA, USA). The ERCC RNA Spike-In Mix (ThermoFisher Scientific, Waltham, MA, USA) was added to normalize total RNA before library preparation, following the manufacturer’s protocol.

RNA sequencing libraries were prepared using the NEBNext Ultra II RNA Library Preparation Kit for Illumina, following the manufacturer’s instructions (New England Biolabs, Ipswich, MA, USA). Briefly, mRNAs were initially enriched with Oligo d(T) beads. The enriched mRNAs were fragmented for 15 min at 94 °C. Subsequently, first-strand and second-strand cDNAs were synthesized. The cDNA fragments were end-repaired and adenylated at 3′ends, and universal adapters were ligated to the cDNA fragments, followed by index addition and library enrichment by PCR with limited cycles. The sequencing libraries were validated using the Agilent TapeStation (Agilent Technologies, Palo Alto, CA, USA) and quantified using the Qubit 2.0 Fluorometer (ThermoFisher Scientific, Waltham, MA, USA) as well as by quantitative PCR (KAPA Biosystems, Wilmington, MA, USA).

The sequencing libraries were clustered on one flow-cell lane. After clustering, the flow cell was loaded on the Illumina instrument (4000 or equivalent) according to the manufacturer’s instructions. The samples were sequenced using a 2 × 150 bp paired-end configuration. Image analysis and base calling were conducted by the Control software. Raw sequence data (.bcl files) generated from the sequencer were converted into fastq files and de-multiplexed using Illumina’s bcl2fastq 2.17 software. One mismatch was allowed for index sequence identification. The raw data have been deposited on NCBI under the accession number PRJNA1183571.

### 2.6. Differential Expression Analysis

The quality of raw sequencing reads was analyzed using FastQC [24], version 0.12.1. The adapters were trimmed, and low-quality reads were filtered out using the Trimmomatic program [25] version 0.39. HISAT2 [26] version 2.2.1 was used for alignment to the Gallus gallus genome assembly (GenBank Accession: GCA_016699485.1), sorted binary alignment files were generated using SAMtools [27] version 1.21, and featureCounts [28] (subread module version 2.0.1) was utilized for transcript quantification.

The following steps were performed using RStudio version 2024.04.2+764 [29,30]. Principal component analysis (PCA) was performed using the built-in stats package [30] version 4.4.1 and plotted using ggplot2 [31]. To determine whether the centroids of various groups on PCA plots differed significantly, PERMANOVA (permutational multivariate analysis of variance) [32] was performed using the vegan package [33] version 2.6-10.

Differential expression analysis was performed using edgeR [34] version 4.2.1, and the genes were filtered by log2 fold change (LFC) > 1 and *p* < 0.05, adjusted with the Benjamini–Hochberg correction [35], for false discovery rate (FDR). The counts of DEGs between the control (NC) and the inoculated groups—Low and High S1133 groups—were visualized using stacked bar plots comparing DEGs across time points and tissues. Unique genes in NC vs. Low S1133 and NC vs. High S1133 were analyzed together across time points for each tissue to determine the common DEGs among tissues. The commonalities and set sizes were pictorially represented as an UpSet plot [36], generated using the UpSetR package [37], version 1.4.0. The scripts were slightly modified from a previous analysis (https://github.com/Zubair2021/ARV_Cell_Transcriptomics_2024).

### 2.7. Pathway Annotation and Comparison

The lists of unique DEGs for each tissue obtained from edgeR output were used as the input to compare enriched pathways using Metascape [38] version 3.5.20240901. The pathway networks were visually enhanced using Cytoscape version 3.10.1 [39], and the most significant pathways of the cluster were annotated using AutoAnnotate [40] version 1.3.0.

### 2.8. Protein–Protein Interaction (PPI) Analysis

The interactions of proteins for each tissue were analyzed by inputting the lists of unique DEGs into the STRING database [41]. The clustering patterns of protein interactions were observed for all tissues. The cluster networks were visually enhanced using Cytoscape version 3.10.1 [39].

## 3. Results

### 3.1. Viral Load Analysis

Quantitative PCR targeting the M1 gene of ARV S1133 revealed an overall dose- and time-dependent pattern in the viral loads (Figure 1). Liver samples from embryos inoculated with the high dose of ARV S1133 exhibited the highest viral loads at 24 h post-infection (hpi), although a greater variability in viral load was observed among samples at 48 hpi. Bursae, kidney, and intestinal samples from embryos inoculated with a high dose of S1133 were positive only at 48 hpi with similar levels of viral replication. Among the high-dose samples, there were six positive samples each for the bursa and intestines, while the kidneys and liver had four. Samples from the embryos inoculated with the low dose of S1133 generally exhibited minimal or undetectable viral RNA. Statistical analysis with a three-way ANOVA test confirmed significant effects of treatment and time on viral load (*p* < 0.001 for treatment and *p* = 0.002 for time), with liver samples showing a significantly higher viral load than other tissues.

### 3.2. Principal Component Analysis (PCA) of Gene Expression

PCA analysis of normalized gene counts (Figure 2) demonstrated that tissue type was the primary factor influencing gene expression patterns. Samples from each tissue clustered closely, reflecting intrinsic differences between tissues rather than treatment and time effects. Tissue- and time-specific PCA plots (Figure 3) highlighted distinct clustering of liver samples under control and infected conditions at 24 hpi, suggesting an early transcriptional shift following ARV S1133 infection. The control samples from the kidneys and intestines exhibited greater dispersion along principal component 1 (PC1) compared to infected samples. The bursal samples had greater dispersion along principal component 2 compared to PC1. The spread of data was the least in the Low S1133 group across tissues and time points. The PERMANOVA analysis confirmed tissue type as the only significant source of gene expression variability in the overall matrix (*p* = 0.001). The effect of time post-inoculation (*p* = 0.421) and treatment (*p* = 0.436) remained insignificant.

### 3.3. Differentially Expressed Genes (DEGs) Across Tissues and Timepoints

The number of DEGs varied considerably across tissues, time points, and treatment doses, with notable differences observed upon pooling data across time points (Figure 4).

In the bursa, 17 DEGs were identified at 24 hpi between Low S1133 and NC, and 16 DEGs between High S1133 and NC. The number of DEGs in High S1133 vs. NC increased substantially to 88 by 48 hpi. After pooling the data across time points and groups, the number of unique DEGs in the bursa reached 107, highlighting a delayed but robust transcriptional response under high-dose conditions.

The intestinal samples exhibited a markedly strong transcriptional response, with 993 DEGs for Low S1133 vs. NC and 698 DEGs for High S1133 vs. NC at 24 hpi. At 48 hpi, DEG counts decreased to 297 for Low S1133 and peaked at 557 for High S1133 compared with control. When pooled across time points and groups, the total number of unique DEGs in the intestines was 1471, marking the highest differential expression among the tissues tested.

DEG counts in the kidneys were lower compared to the intestines. At 24 hpi, 278 DEGs were found in Low S1133 vs. NC and one gene in the High vs. Low S1133 comparison. At 48 hpi, 111, 23, and 5 DEGs were identified for High vs. Low S1133, High S1133 vs. control, and Low S1133 vs. control, respectively. Pooling the genes for the High and Low S1133 vs. control comparisons across time points yielded a total of 302 DEGs, indicating a moderate but sustained transcriptional response over time.

The liver displayed an early and robust response, with 210 DEGs for High S1133 vs. NC and 209 DEGs for Low S133 vs. NC at 24 hpi. By 48 hpi, DEGs decreased to 9 for Low S1133 and to 25 for High S1133 in comparison with the control group. After pooling across time points and groups, the liver showed a total of 385 unique DEGs, including some genes directly involved in the inhibition of viral replication. Overall, the liver and intestines contributed the largest unique DEG sets.

The UpSet plot illustrating DEG intersections across tissues (Figure 5) showed the greatest intersection between the liver and the intestines, with 90 common genes, followed by the intestines and the kidneys, sharing 58 genes. The intestines and the bursa had 25 DEGs in common. Only two DEGs were found to be common across all tissues. Of these, LOC121110849 was identified as a 28S ribosomal RNA gene and LOC121107011 as an uncharacterized locus.

### 3.4. Protein–Protein Interaction (PPI) Network and Functional Enrichment Analyses

The PPI network (Figure 6) constructed for liver-specific DEGs identified several antiviral genes as central nodes, including IFI6, RSAD2, MX1, CMPK2, and OASL. The Gene Ontology database pathway enrichment analysis identified several immunity-related pathways in liver samples. Figure 7 highlights pathways directly involved in antiviral defense, such as interferon signaling in the liver, and those indirectly linked to immunity, including blood coagulation and complement cascades in other tissues. Unexpectedly, bursal samples had responses including ribosomal subunit assembly and DNA recombination.

The Metascape network analysis (Figure 8) showed complement systems and pathways related to coronavirus disease at the center of the network. While the pathways were interconnected for various tissues, the clustering of pathways based on tissue type was masked by a significantly larger number of DEGs identified in intestinal samples. The pie charts at the nodes were dominated by intestinal DEGs, masking the contribution of genes from other tissues.

Figure 9 presents additional tissue-specific pathways identified by STRING database PPI clustering. Briefly, an enrichment of clotting cascade and fibrin-related pathways in the intestines and kidneys, protease- and peptidase-associated activity in the bursa, and negative regulation of viral genome replication in the liver samples were observed.

## 4. Discussion

The objective of the study was to ascertain the molecular responses of various chicken embryonic tissues to ARV inoculation. A virulent isolate of ARV S1133 [42] was used for this purpose, the experimental vertical transmission of which had been demonstrated earlier [43]. Due to the relevance of late-stage embryos for in ovo vaccination [12,13,14], and continued maturation of their immune organs [8,9,44], we inoculated SPF embryos at 18 DOE with two different doses of ARV S1133 and sampled various tissues at 24 and 48 hpi.

Following inoculation of the higher dose of ARV S1133, samples from the bursae, intestines, and kidneys indicated a delayed replication of ARV in these organs as they were only positive at 48 hpi, with bursae and intestines having the highest number of positives. In adult chickens, earlier studies suggested a quick (around 6 hpi) viremic spread to the liver following oral ARV inoculation at a similar dose, albeit at a lower infectious titer than the duodenum [45]. In the current study, the liver was found to be the only ARV-positive tissue in the high S1133 group at 24 hpi, displaying the highest viral loads and suggesting a comparatively earlier but more robust replication in this organ. This finding is consistent with our observation in cultured primary liver cells (CELi), where the titers were higher compared to primary chicken embryo kidney cells (CEK) [46]. Moreover, substantial evidence from earlier studies designated the liver as an important organ for examining in ovo pathogenicity, as it exhibits a greenish appearance and necrotic lesions inoculated with ARV at an earlier DOE [6,16,47,48]. While it is difficult to inoculate via yolk sac at 18 DOE, different infection outcomes could have been observed with yolk sac inoculation at an earlier DOE [5]. Notably, while the same dose (100 µL of 10^6^ TCID50) of the ARV S1133 has previously resulted in deaths of embryos when inoculated via yolk sac at 7 DOE in our laboratory, the lesions or mortalities were not noticed in the current experiment. Such an observation could be attributed to the recruitment of immune cells at this later stage of development, as the induction of macrophages has been demonstrated in chicken embryo livers after 12 days of embryonic development (DOE) [49]. Additionally, monocytes and heterophils remain relevant in the livers of adult chickens infected with ARV S1133, as determined by histopathological examination [50,51]. However, the short period of incubation (48 hpi) in this study did not allow for conclusions to be drawn about the adaptive immune responses.

A tissue-dependent clustering on a PCA plot of normalized DEG counts suggested unique expression profiles of embryonic tissues at 19 and 20 DOE, suggesting sufficient differentiation of tissues at this age and distinct molecular mechanisms. A vast variation among the tissues masked the effect of treatment and time points in the overall PCA plot. This suppression of less variable subgroups by larger variable groups is a key limitation of pooled PCA analysis [52]. Interestingly, samples from livers clustered at a greater distance compared to the other tissues in the overall plot. When visualized on the 24-hpi-specific plot, the effect of infection was spatially apparent in the liver samples. Moreover, a slightly lower overall dispersion of infected samples compared to controls on timepoint-specific PCAs for kidneys and intestines indicated a more infection-directed gene expression pattern. Taken together, while PCA provided some insights into the broader clustering patterns of the expression data, the limitations of the tool [53] obscured a holistic picture of differential expression across variables.

The differential expression analysis revealed a varying magnitude of responses across tissues, with particularly notable numbers of DEGs for the intestines, where more than 1000 DEGs were identified at both time points for various comparisons. This observation was concordant with a higher number of ARV-positive intestinal samples at 48 hpi compared to the liver and kidneys. While pre-hatch intestinal replication of ARV following inoculation at 18 DOE has never been explored previously, the virus has been isolated from the intestines of three-day-old commercial broiler chicks [54], as well as chicks hatched infected through experimental vertical transmission [55] or infected as embryos [48]. In the current study, despite a greater number of positive intestinal samples, the relative viral RNA measurements were similar to those of kidneys and bursae. This disparity between viral load and the magnitude of transcriptional response suggested that the intestinal tissue might be implementing a pre-emptive defense strategy to the viral presence. Previously, researchers investigating avian gut granulopoiesis suggested the possible migration of immature heterophils to the intestines before hatch and retention of their capability to divide outside the bone marrow [56], implying the potential of the intestines to mount relevant immune responses pre-hatch and serve as a primary barrier tissue upon hatch. It is important to acknowledge that the pre-hatch and post-hatch host responses may differ significantly due to maturation of the immune system [8,9].

While ARV has frequently been isolated from cecal tonsils [1] and is known to cause bursal follicular atrophy [57] and lymphoid depletion [58], the antiviral immune responses of the lymphoid tissues following embryonic ARV inoculation have not been characterized. In the present study, bursae showed the least number of DEGs among the tissues tested. A higher number of ARV-positive bursal samples with low relative viral RNA measurements hint at a potential lack of active replication and might be attributed to the anal sucking-like movements in embryos, a phenomenon that takes up in ovo trace materials into the bursal lumen after 15 DOE [59]. Moreover, the pathways enriched in bursal samples were neither similar to those enriched in the intestines, nor relevant to antiviral responses. A previous report has indicated the development of Peyer’s patches and cecal tonsils independent of the bursa with the accretion of major histocompatibility complex class-II-positive cells as early as 13 DOE [60], reaffirming the intestines’ ability to induce relevant defense responses when the bursa is not fully developed.

The pathways enriched in the intestines and kidneys were predominantly blood-related, i.e., linked to coagulation or complement cascades. These results were in accordance with the findings in CEK cells [46]. While the explanations for the direct role of these blood-related pathways remain indefinite, an induction of angiogenesis by macrophages [61] could partially explain the indirect upregulation of these pathways as a consequence of viral infection. Previously, researchers have made similar observations with infectious bronchitis, where DEGs related to complement factors and blood vessel-associated pathways were identified [62]. Moreover, the complement system has been shown to contribute to immune responses against influenza virus [63], human immunodeficiency virus [64], Sindbis virus [65], and dengue virus [66] infections. The network analysis also revealed an unexpected clustering of pathways related to coronavirus disease, highlighting potential commonalities in host responses to different viral families. Such an observation could be attributed to a limitation of the analysis, as the Metascape database utilizes human orthologs of the DEGs.

An enrichment of several antiviral Gene Ontology pathways and significant PPI in the liver tissue marked an orchestrated cellular response to inhibit viral replication. The differential enrichment of PPI encoded by genes such as IFI6, RSAD2, IFIT5, MX1, USP41, ZNFX1, and OASL in embryonic liver tissue is consistent with our findings on CELi cells (unpublished data). However, the network for the tissue included more genes compared to cultured CELi cells. Nonetheless, the protagonism of some of these genes for core antiviral response has been demonstrated against avian reovirus in vivo [20,67,68] and in vitro [19], as well as against other avian viruses [69,70,71,72,73,74].

In conclusion, the study provides insights into the systematic responses of four distinct tissue types at two different timepoints following ARV infection. It can be concluded that ARV replication efficiency and the virus-induced host transcriptional landscapes are dose-, time- and tissue-dependent, and a variety of metabolic pathways are enriched as a response to in ovo ARV inoculation.

## Figures and Tables

**Figure 1 viruses-17-00646-f001:**
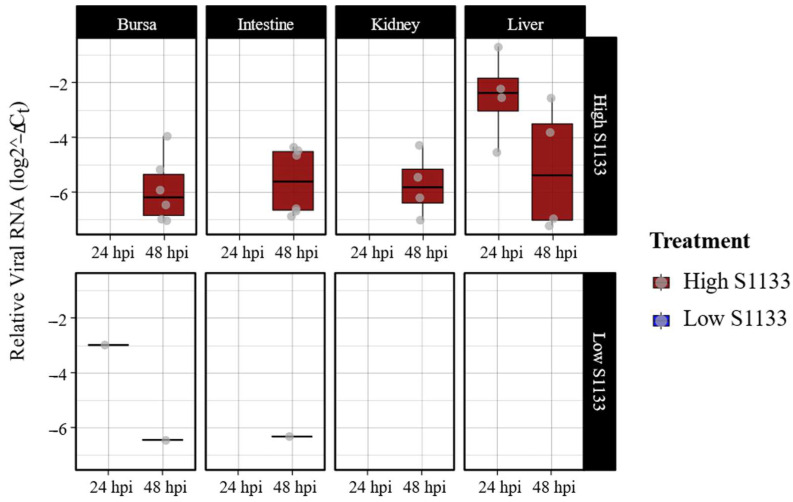
Avian reovirus S1133 replication in various embryonic tissues over time (*n* = 7). The x-axes indicate the sampling timepoint in hours post-inoculation (hpi). The y-axes represent relative viral RNA calculated using threshold cycle values (Ct) of qPCR targeting ARV M1 gene and normalized against housekeeping gene GAPDH RNA using the formula log (2^−(GAPDH Ct−ARV Ct)^). Low S1133 samples were predominantly negative, with a few exceptions. The figure only shows data for positive samples.

**Figure 2 viruses-17-00646-f002:**
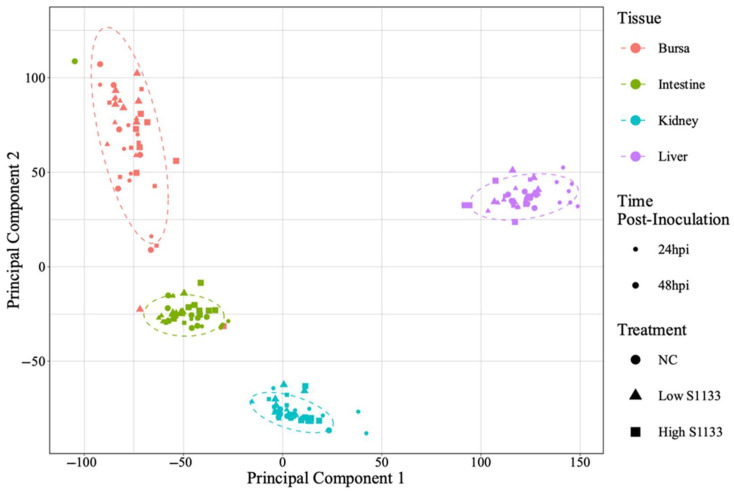
Overall principal component analysis (PCA) of normalized gene count distribution for various tissues (*n* = 7). The shapes of different colors represent samples from tissue types collected from the specific-pathogen-free embryos. The treatment groups, including the control (NC) and the low and high doses of ARV S1133, are represented by the shapes. The size of the points represents time as hours post-inoculation (hpi). The variance along the x- and y-axes, represented by principal components 1 and 2, primarily reflects the influence of tissue type on gene expression profiles across samples. Within each tissue type, samples clustered closely together irrespective of time point or treatment, suggesting a high degree of similarity in their transcriptional profiles.

**Figure 3 viruses-17-00646-f003:**
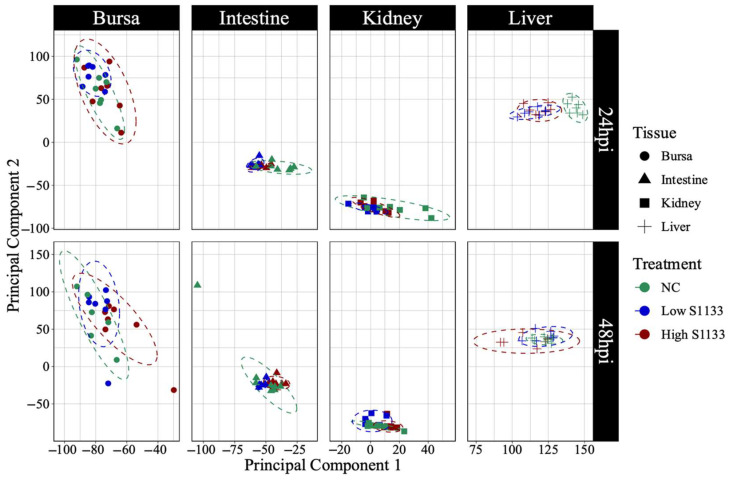
Principal component analysis (PCA) of normalized gene count distribution by tissue and time (*n* = 7). The shapes represent samples of various tissues collected from specific pathogen-free embryos 19 and 20 days of embryonation. The samples from the control (NC) and the low and high doses of ARV S1133 groups are represented by colors. The control liver samples at 24 h post-inoculation (hpi) clustered distinctly compared to infected ones. While samples from the intestines and kidneys had greater dispersion along the x-axis (principal component 1), those from bursae had more vertical distribution along principal component 2.

**Figure 4 viruses-17-00646-f004:**
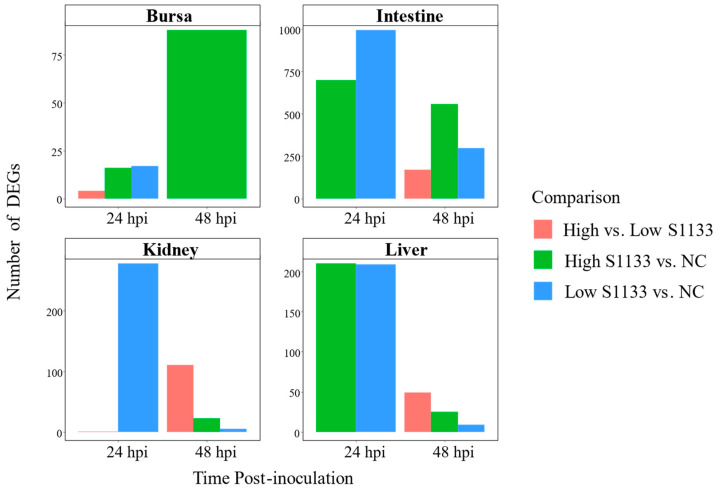
Counts of differentially expressed genes (DEGs) across embryo tissues and time points following avian reovirus (ARV) S1133 inoculation (*n* = 7). The bar plots compare the number of DEGs identified for comparisons involving DMEM-inoculated control (NC) or low and high doses of ARV S1133. The x-axes represent hours post-inoculation (hpi) and the y-axes indicate the counts of DEGs for each contrast. The highest numbers of DEGs were identified in the intestines and liver at 24 hpi. Except for bursa samples, a decline in the total number of DEGs was observed for each tissue over time. To help visualize smaller differences, the y-axes are scaled independently for each tissue.

**Figure 5 viruses-17-00646-f005:**
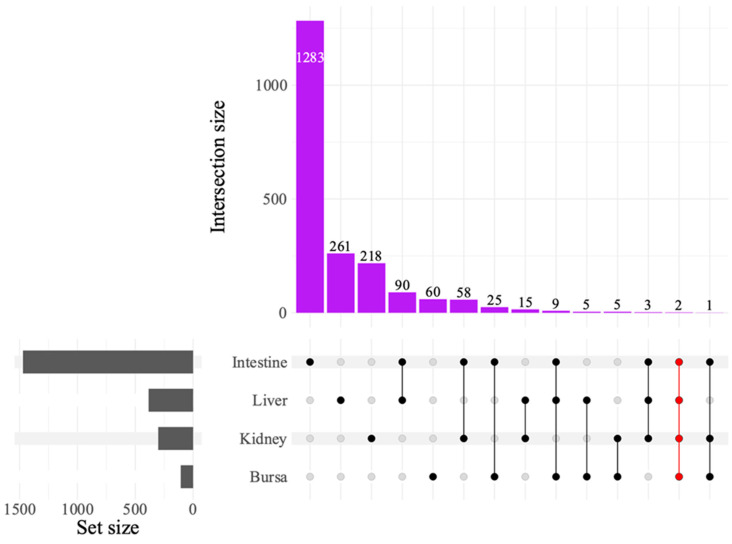
UpSet plot of differentially expressed genes (DEGs) between avian reovirus (ARV) S1133 strain-infected and control samples in various tissues (*n* = 7). The grey bars represent the number of DEGs observed in samples from various tissues of specific-pathogen-free embryos inoculated with ARV S1133 or DMEM. The black and red dots connected by black or red lines on the x-axis indicate the comparison between respective tissue types for which common DEGs were identified. The DEGs from both time points and comparing high and low doses of ARV S1133 were pooled for each tissue. The numbers above the purple bars represent the number of DEGs for the respective intersections. Only two common DEGs were identified as common across all four tissues tested, indicating an organ-specific transcriptional response.

**Figure 6 viruses-17-00646-f006:**
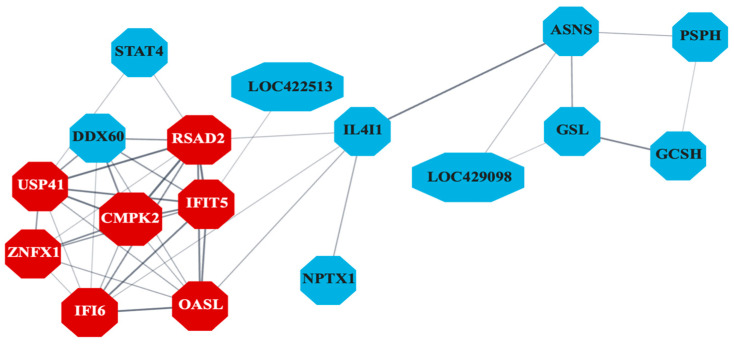
Protein–protein interaction (PPI) network indicating co-expression of differentially expressed genes (DEGs) related to antiviral activity in liver samples (*n* = 7). Genes differentially expressed between control and S1133-infected cells in chicken embryo liver samples 24 and 48 h post-inoculation were analyzed to create a PPI network. The network reveals significant interactions among genes involved in the inhibition of viral replication with key genes such as IFI6, RSAD2, IFIT5, MX1, USP41, ZNFX1, and OASL (shown in red), and other related genes, shown in blue, highlighting their importance in mediating the cellular response to ARV S1133 infection. Only the network with the highest number of interactions in the STRING database is shown here.

**Figure 7 viruses-17-00646-f007:**
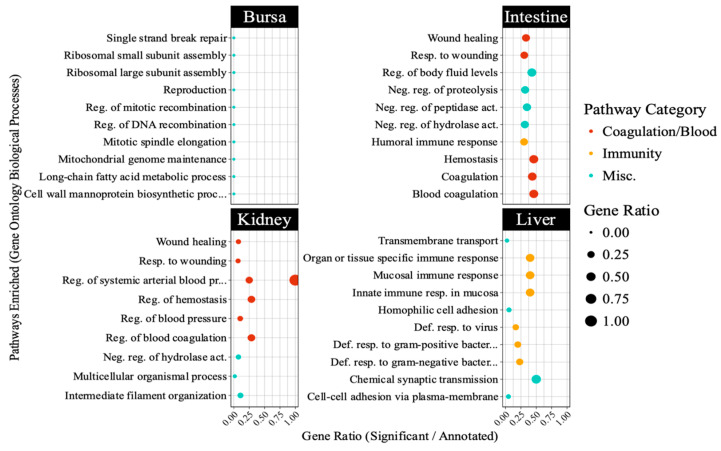
Top 10 pathways enriched in different embryonic tissues following avian reovirus (ARV) S1133 inoculation (*n* = 7). The table indicates pathways representing gene ontology (GO) database terms enriched in each tissue following ARV S1133 inoculation. The x-axes and size of the dots indicate the ratio of differentially expressed genes (DEGs) vs. the total number of genes annotated in the database for a given pathway. The y-axes indicate categories of Gene Ontology database-retrieved pathways enriched in the respective tissues. The orange-colored dots, predominantly enriched in the liver, indicate pathways related to immunity, and those in red depict pathways related to blood, coagulation, or complement system. Abbreviations: “Def. = “defense”, “reg. = regulation”, “resp. = response”, “Misc. = miscellaneous”.

**Figure 8 viruses-17-00646-f008:**
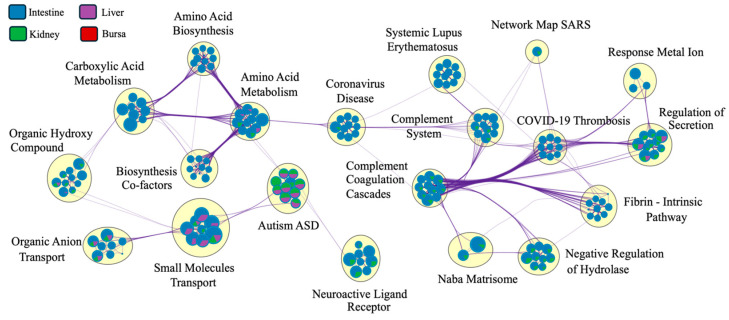
Pathway connectedness network of differentially expressed genes (DEGs) from each tissue (*n* = 7). Each pie chart represents a node, where each node corresponds to a cluster of DEGs (mapped against the human gene ontology database in Metascape). Nodes are linked by edges. A higher number of edges connecting clusters indicates greater connectedness. The size of each node is proportional to the number of DEGs within the cluster. The big yellow circles indicate clusters of closely related pathways, highlighting functionally similar or overlapping biological processes (only the most significant pathway is labeled here). The colors within each pie chart display the percentage of DEGs derived from each tissue (legend on the top left). Since the number of genes for the intestines is significantly higher, the network was dominated by pathways related to DEGs expressed in the intestines.

**Figure 9 viruses-17-00646-f009:**
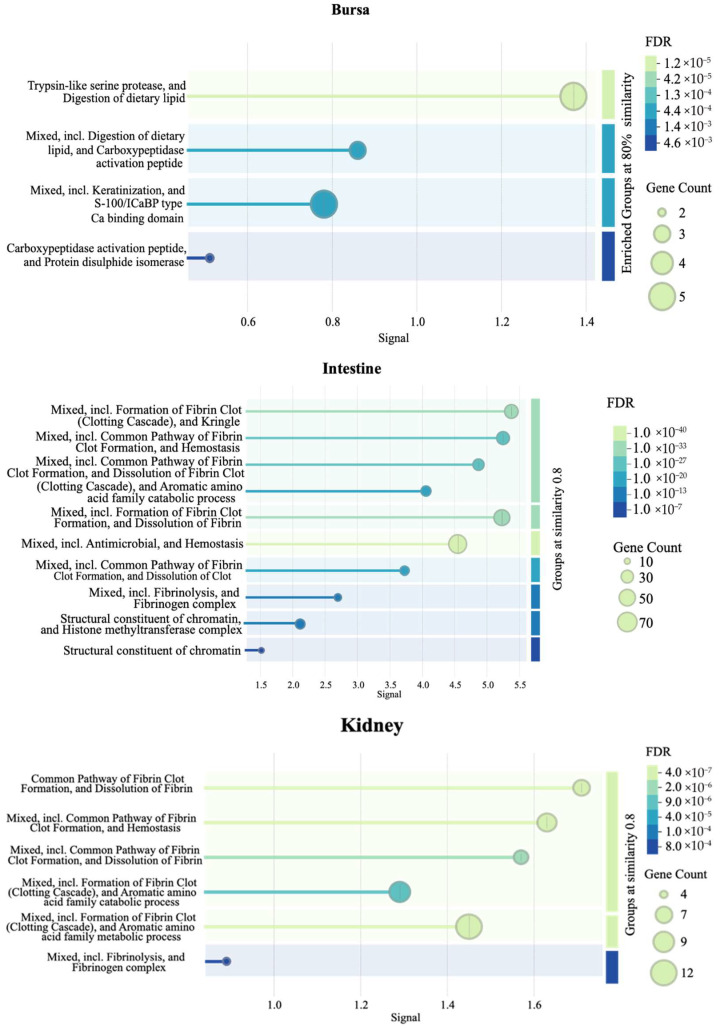
Pathways enriched in each tissue as identified by a protein–protein interaction (PPI) network (*n* = 7). The PPI network was generated using the STRING database to visualize the interactions among significant DEGs in each tissue. The signal values on the x-axes reflect the strength or confidence of the interaction. Node size represents the gene count associated with a pathway, while the color gradient of the dots reflects the significance of the false discovery rate (FDR).

## Data Availability

The raw reads for each sample are available at GenBank (Accession: PRJNA1183571).

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
