# Peer review of "Tissue-Specific Transcriptomic Responses to Avian Reovirus Inoculation in Ovo"

_viruses, 2025, doi:10.3390/v17050646_

Round 1
Reviewer 1 Report
Comments and Suggestions for Authors
Avian reovirus (ARV) infections significantly impact the global poultry industry, but host responses across infection models remain poorly characterized. This study examined tissue-specific transcriptomic changes following in ovo inoculation with two doses of ARV S1133 at embryonic day 18. Quantitative PCR confirmed that different tissues exhibiting different viral loads at different time points. Further transcriptomic profiling revealed an extensive gene expression response in the intestine, implicating early immune activation. Liver samples demonstrated strong upregulation of antiviral pathways, including interferon signaling and viral replication inhibition, while kidneys and intestines were enriched for coagulation and wound healing pathways. The bursae exhibited minimal immunity-related responses, suggesting insufficient maturation. Functional analyses confirmed tissue-specific immune and metabolic adaptations to infection. These findings indicate that ARV replication efficiency and host molecular responses are dose-, tissue-, and time-dependent. This study establishes foundational knowledge of host molecular responses to ARV in late-stage embryos, with implications for in ovo vaccination and early immunity. This study is somewhat innovative, but there are still some problems to revise.
- The study used embryo injection to inoculate ARV, while in natural infections, poultry are mainly infected ARV through the digestive tract or respiratory tract. Will different inoculation routes lead to deviations in virus distribution and host response? Please provide additional explanations in the discussion section.
- 18-day-old embryos (close to hatching) are selected in this study, but the text also mentions that the chick immune system continues to mature after hatching. Whether the host's antiviral response of not fully mature T cells and B cells of 18-day-old embryos can reflect the true host response.
- In this study, the control group was inoculated with DMEM. Why didn’t you set up an un-inoculated blank control group. You know that, such as mechanical damage, the injection procedure may potentially activate a non-specific stress response.
- The liver and intestine exhibited completely different transcriptional responses (e.g., strong antiviral pathways in the liver vs. coagulation and wound healing pathways in the intestine). Is this difference related to the cellular composition of the tissues themselves (e.g., hepatocytes vs. intestinal epithelial cells/immune cells)?
- The study primarily focused on the changes in the transcriptome, but did not further explore the molecular mechanisms underlying these changes. A large number of differentially expressed genes (DEGs) were identified in the study. It is recommended to supplement the study with functional validation experiments for these genes (such as qPCR or Western blot) to ensure the reliability of the transcriptome data.
Author Response
Reviewer 1
Comments and Suggestions for Authors
- The study used embryo injection to inoculate ARV, while in natural infections, poultry are mainly infected with ARV through the digestive tract or respiratory tract. Will different inoculation routes lead to deviations in virus distribution and host response? Please provide additional explanations in the discussion section.
Response: Thank you for reviewing the manuscript. We have addressed this in the discussion as suggested (lines 350-351).
- 18-day-old embryos (close to hatching) are selected in this study, but the text also mentions that the chick immune system continues to mature after hatching. Whether the host's antiviral response of not fully mature T cells and B cells of 18-day-old embryos can reflect the true host response.
Response: While we acknowledge the limitations, we used embryo as an infection model possibly closest to a hatched chick. As mentioned in the introduction, later time points or 18 DOE are better suited to investigate the molecular responses to viral infections. The immune system continues to mature after hatch and therefore, the perfect time to study host response during the early age of the chick is not known. Moreover, the antiviral responses at 18 DOE are of significant relevance for in ovo immunization against ARV. The administration of a commercial ARV vaccine in ovo has been shown to suppress post-hatch T cell-mediated immunity, but the molecular responses and mechanisms causing immunosuppression are unexplored. While the transcriptome-wide gene expression patterns of fibroblast cells and spleen from SPF chickens infected with ARV have been studied, the embryos’ organ-specific molecular response to ARV inoculation remains unexplored. Hence, the objective of this study was to explore the transcriptomic changes occurring in embryos following ARV infection that may cause immunosuppression post-hatch.
We now point out the limitation of the difference between the immune system pre- and post-hatch (lines 391).
- In this study, the control group was inoculated with DMEM. Why didn’t you set up an un-inoculated blank control group? You know that, such as mechanical damage, the injection procedure may potentially activate a non-specific stress response.
Response: The normal chicken embryo transcriptome has already been described and was beyond the scope the study. The objective of this study was to compare the transcriptional responses between ARV-inoculated and sham-inoculated embryos. To ensure a fair and accurate statistical analysis, it was essential to account for any transcriptomic changes that could arise from the injection procedure itself, including potential stress responses due to mechanical damage. Therefore, instead of using an uninoculated blank control, we included a DMEM-inoculated control group. Since DMEM was the medium used to culture and deliver the virus in the infected group, this approach allowed us to assess the effects of viral infection specifically while controlling for any non-specific responses triggered by the injection process.
- The liver and intestine exhibited completely different transcriptional responses (e.g., strong antiviral pathways in the liver vs. coagulation and wound healing pathways in the intestine). Is this difference related to the cellular composition of the tissues themselves (e.g., hepatocytes vs. intestinal epithelial cells/immune cells)?
Response: The observed differences in transcriptional responses between the liver and intestine are likely influenced by the distinct cellular composition and functional roles of these tissues. The liver, primarily composed of hepatocytes along with Kupffer cells and other resident immune cells, plays a central role in metabolism and systemic immune responses, which could explain the strong activation of antiviral pathways. In contrast, the intestine, which consists of epithelial cells, immune cells, and fibroblasts, is critical for maintaining barrier integrity and wound healing. The upregulation of coagulation and wound healing pathways in the intestine suggests a localized response to tissue damage or immune activation.
- The study primarily focused on the changes in the transcriptome, but did not further explore the molecular mechanisms underlying these changes. A large number of differentially expressed genes (DEGs) were identified in the study. It is recommended to supplement the study with functional validation experiments for these genes (such as qPCR or Western blot) to ensure the reliability of the transcriptome data.
Response: Thank you for your suggestion. We agree that qPCR or western blot validation of selected DEGs could further strengthen our findings, and we acknowledge this as a valuable next step for future studies. However, for this hypothesis-generating study, we primarily aimed to provide a comprehensive overview of host transcriptomic response, identifying key differentially expressed genes (DEGs) and pathways involved. To ensure data reliability, we employed stringent statistical thresholds and quality control measures during analysis. In fact, as indicated by the several analyses, the core genes identified in our study have already been validated by several other experiments with avian viruses.
Reviewer 2 Report
Comments and Suggestions for Authors
Minor points :
- Line 16-18: please describe the specific pathways related with the antiviral transduction and replication.
- Line 77-78: please describe the inoculation route into embryo.
- Line 100-102: the annealing temperature of primers and recycling parameters of qPCR were required.
- Line 220-221: it would be better if all the Y axis of figure 4 adjusted to the same scale.
- Line 437/439: the format of reference needs to be consistent.
Major points:
- Line 75-76: why only two doses were evaluated and analyzed, it would be better if there were another dose group, such as 104 TCID50/mL.
- Line 171-173/237-239/376-378: The number of unique DEGs is the highest in the intestine, but relatively lower viral load was found. There is no direct evidence of immune response behind the findings.
- Line 337-339: it would be better if the adult chickens infected with ARV were included in the study.
no comments
Author Response
Reviewer 2:
Minor points :
- Line 16-18: please describe the specific pathways related with the antiviral transduction and replication.
Response: We added the specific pathways in Lines 16-18: Liver samples demonstrated strong upregulation of antiviral pathways, including interferon signaling and viral replication inhibition, while kidneys and intestines were enriched for coagulation and wound healing pathways.
- Line 77-78: please describe the inoculation route into embryo.
Response: We added this as suggested (line 77-78).
- Line 100-102: the annealing temperature of primers and recycling parameters of qPCR were required.
Response: We included this information (lines 99 – 104).
- Line 220-221: it would be better if all the Y axis of figure 4 adjusted to the same scale.
Response: We acknowledge that theoretically this would be very much preferable. However, using a uniform Y-axis scale across all graphs would obscure subtle differences in the groups with smaller ranges. To ensure that these changes remain visible and interpretable, we chose to scale each graph individually.
- Line 437/439: the format of reference needs to be consistent.
Response: The first reference is an online journal, so the article number has been provided and the second contains the page numbers.
Major points:
- Line 75-76: why only two doses were evaluated and analyzed, it would be better if there were another dose group, such as 104 TCID50/mL.
Response: We agree that it would always be desirable to include more groups and tests, but we could not do that due to logistic and financial restraints. We used 100uL of 104 TCID50/mL as it has been previously used in studies on ARV vaccination. We included a relatively higher dose to determine if there will be a dose-based response on transcriptome in embryos.
- Line 171-173/237-239/376-378: The number of unique DEGs is the highest in the intestine, but relatively lower viral load was found. There is no direct evidence of immune response behind the findings.
Response: We agree that the lower viral replication in the jejunum with higher DEGs seems paradoxical but is in fact not entirely unexpected. As discussed in lines 384-386, a lower viral load but a higher transcriptional response may indicate that the respective organs were able to mount a significant immune response to curtail viral replication. As pointed out by the reviewer, a lack of DEGs indicating an immune response in the jejunum indicates an ineffective immune response allowing more virus replication.
Bursa and kidneys had no detectable viral loads at 24 hpi, which indicates an ineffective viral replication and would explain the lower number of DEGs.
In contrast. the genes identified in the protein-protein interaction network in the liver showed a direct connection with inhibition of viral replication as these proteins constituted core antiviral response. This is suggestive that the immune responses were indeed elicited and would explain the comparatively lower viral titers in the liver.
- Line 337-339: it would be better if the adult chickens infected with ARV were included in the study.
Response: As indicated above, we agree that it would always be better to add more trials, however unfortunately there are logistic and financial restraints. This study used late stage embryos as a feasible alternative to a complete animal trial.
Reviewer 3 Report
Comments and Suggestions for Authors
Although the authors made a good attempt to study the gene expression in different organs following ARV inoculation in late embryo, but it is not clear what did the authors wanted to present in this manuscript such as-
- It is not clear what was the aim of the study?
- What was the idea behind studying the gene expression following low and high dose ARV injection? instead a similar study could have been conducted following inoculation of a non-pathogenic and a pathogenic ARV strain. Or inoculation of a vaccine strain to identify the genes if any related to immuno suppression.
- How did the authors compared the DEG enriched pathways using Metascape? currently Metascape only support 10 species and chicken is not included.
- Figure 2 and 3 is not adding anything to the paper
- Figure 4 can be added to supplemental data.
- There are other comments which are in the attached file.

Author Response
Reviewer 3:
Although the authors made a good attempt to study the gene expression in different organs following ARV inoculation in late embryo, but it is not clear what did the authors wanted to present in this manuscript such as-
- It is not clear what was the aim of the study?
Response: The aim has been rewritten to make it clear as suggested (lines 59-62).
- What was the idea behind studying the gene expression following low and high dose ARV injection? instead a similar study could have been conducted following inoculation of a non-pathogenic and a pathogenic ARV strain. Or inoculation of a vaccine strain to identify the genes if any related to immuno suppression.
Response: We appreciate the reviewer’s insightful suggestion of comparing gene expression following inoculation with non-pathogenic versus pathogenic ARV strains or vaccine strains. Such comparisons could provide valuable information on virulence-associated or immunosuppressive gene signatures, and we might do these in the future. However, our current study was designed with a different but complementary goal — to investigate the dose-dependent effects of a well-characterized pathogenic ARV strain (S1133) on the transcriptomics of embryonic tissues at a critical stage of immune development (18 DOE). The use of low and high viral doses aimed to mimic varying levels of vertical transmission or field exposure and to differentiate threshold-dependent host responses, particularly those that might contribute to immunosuppression.
- How did the authors compare the DEG enriched pathways using Metascape? currently Metascape only support 10 species and chicken is not included.
Response: We are aware that Metascape doesn’t have a chicken-specific database. For this reason, we used the human gene ontology database (Line 310). However, as mentioned on the website (https://metascape.org/gp/Content/MenuPages/faq.html): “By default, all input gene identifiers are converted into their human orthologs for annotation and enrichment analysis, as the knowledge base for the humans is the most comprehensive.” We acknowledge it as one of the limitations of the analysis (Lines 417- 419). However, the use of the database across species is not a novel practice.
- Figure 2 and 3 is not adding anything to the paper
- Figure 4 can be added to supplemental data.
Response: Figure 2 is included to show that tissue type was the primary factor that determined the gene expression profile irrespective of the time of sample collection and the virus concentration. Figure 3 is included to show that for each tissue and infection dose there is a limited influence of the infection status. We ask the reviewer to respect our preference to show more information if in doubt.
- There are other comments which are in the attached file.
Reviewer 3:
Line 18-19: Our findings align with this understanding, as we observed minimal immunity-related responses in the bursae, suggesting insufficient maturation.
Response: We have now clarified this in the manuscript to reflect the expected timeline of humoral immune development (lines 37-41).
Line 68: Please revise the virus titer. From this titer how was 10^4 & 10^6 calculated in the next line?
Response: Corrected the titer as suggested. This was a mistake made while copying the manuscript to the template- the superscript was missing (line 67-70).
Line 76: Comparing with other available literature, this seems to be very high dose for inoculating a pathogenic strain. What is the idea behind using the pathogenic strain for in ovo, if you wanna apply similar work to references 10 and 11 that you mentioned in your introduction? I think it will be better to use low pathogenic form if you wanna check the changes or suitability for in ovo vaccination.
Response: In the referenced studies, they used EID50 as a measure of viral concentration. But in this study, we used TCID50 for consistency between our in vitro, in ovo, and in vivo studies.
The dose used is according to the previous studies on ARV vaccination (https://doi.org/10.1080/03079457.2024.2425353). We included a higher dose to study the dose-dependent responses to ARV infection.
Furthermore, the lack of gross lesions in the embryos confirms that the titer was not too high.
Line 77: After puncturing the eggshell, the inoculum was injected into the embryo using a one-inch 21-gauge needle fully inserted at a 45-degree angle.
Which route?
Response: The inoculum was deposited in the allantoic fluid. We added this to the materials and methods as suggested (line 77-78).
Line 83: No mortality was observed in any group after the injection of the respective inoculum.
Do you mean that there was no mortality observed after 2 days using 10^6 TCID50 of the pathogenic strain S1133?
Response: This is correct, no mortality was observed after inoculating the viruses -24 hours and 48 hours post-infection. This should not be a surprise. Inoculation into the allantoic fluid is similar to oral infection, which does not cause acute deaths in young chicks. Obviously, the late stage embryos are much more similar to freshly hatched chicks than they are to 10 day old embryos, in which the dose would cause mortality.
Line 92: Total RNA was denatured for 10 minutes at 95ËšC -Too much time for RNA denaturation
Response: This was an error. It has been corrected in the manuscript as suggested (line 91).
Line 157-158: The lists of unique DEGs for each tissue obtained from edgeR output were used as input to compare enriched pathways using Metascape [38] version 3.5.20240901.
Metascape primarily supports human and model organisms like mouse and rats. But Chicken is not fully supported. So have you converted the chicken gene IDs to human orthologs (Because it has not been mention or there is no reference for it)?. Others such as BioMart (Ensembl) could have been used to convert chicken Ensembl IDs or gene symbols to human orthologs and you can also use NCBI HomoloGene or OrthoDB for curated orthology data. Then you have to prepare DEGs list via organizing ADEGs for each tissue or do a comparison in separate gene lists and then use human orthologs for Metascape submission.
Response: For this comment, we refer to our reply to the reviewer’s similar comment above.
Line 175-176: Samples from the embryos inoculated with the low dose of S1133 generally exhibited minimal or undetectable viral RNA.
What is the qPCR LoD and LoQ for your qPCR? Have you defined it using serial dilutions? This is helpful to know if the virus is not replicating or the method is not robust enough to reveal the low level of virus replication, especially during this short window.
Response: We have not determined the LoD or LoQ for our qPCR but as indicated by the time post-inoculation trends, the virus did replicate to detectable levels in all the organs tested at 48 hpi with the higher dose (except for liver, where it was also detected at 24 hpi). In any case, replication of the virus at levels below the LoD is the most likely explanation for the observed changes in the transcriptome.
Line 275-276: Interestingly, bursal samples had responses including ribosomal subunit assembly and DNA recombination.
What could be interesting about the bursa responses to ARV infection?
Response: We have rephrased the sentence (lines 283-284).
Line 298: The Metascape network analysis (Figure 8) showed complement systems and pathways related to coronavirus disease at the center of the network.
Have you used the Orthologs in your analysis?
Response: For this comment, we refer to our reply to the reviewer’s similar comment above.
Line 328-329: The objective of the study was to ascertain the molecular responses of various chicken embryonic tissues to ARV inoculation
In any viral infection we are going to see some changes in the tissue/organ where it replicates?, what does it mean?
Response: We agree with the reviewer that there should be molecular changes in organs/tissues where the virus replicates. However, these changes have not been characterized in a chicken embryo near hatch following ARV infection and we conducted this study to characterize those changes.
Line 338: This is in chicks which could be entirely different than embryonic eggs and especially when inoculation is at DOE 18.
In adult chickens, earlier studies suggested a quick, about ( 6 hpi, viremic spread to the liver following oral ARV inoculation at a similar dose, albeit at a lower infectious titer than duodenum.
Response: Since there are no studies comparing the spread of the virus at 18 DOE among organs, we substantiated the discussion with a relevant piece of information from a study using adult chicken (line 341-343).
Line 349: 350: Do you mean that the innate immune response will be enough to prevent mortality when innoculated with such high dose? because the adaptive responses (e.g. antibodies or T-cell-mediated responses) probably won't be evident in this short window, especially pre-hatch.
Such an observation could be attributed to the recruitment of immune cells at this later DOE, since the induction of macrophages has been shown in chicken embryo liver after 12 DOE [49] and monocytes and heterophils remain relevant in livers of adult chickens infected with ARV S1133, as determined histopathologically [50,51].
Response: Thank you for pointing it out. We have clarified it in lines 359-361. Please consider also our comment regarding the infection dose above.
Round 2
Reviewer 1 Report
Comments and Suggestions for Authors The manuscript has been sufficiently improved to warrant publication in Viruses.Reviewer 3 Report
Comments and Suggestions for Authors
I think the authors address all the queries